# Effect of Two Strength Training Models on Muscle Power and Strength in Elite Women’s Football Players

**DOI:** 10.3390/sports8040042

**Published:** 2020-03-30

**Authors:** Martin Pacholek, Erika Zemková

**Affiliations:** 1Physical Education Department, College of Applied and Supporting Studies, King Fahd University of Petroleum & Minerals, Dhahran 31261, Saudi Arabia; 2Department of Biological and Medical Sciences, Faculty of Physical Education and Sports, Comenius University in Bratislava, 81469 Bratislava, Slovakia; erika.zemkova@uniba.sk; 3Sports Technology Institute, Faculty of Electrical Engineering and Information Technology, Slovak University of Technology in Bratislava, 81219 Bratislava, Slovakia

**Keywords:** combined and complex model, training methods, women’s soccer, performance

## Abstract

This study evaluates changes in power and strength after implementing two different models of 9-week strength training in elite women’s football players. A group of 13 players (age 20.2 ± 3.3 years, body mass 57.2 ± 3.7 kg, height 163.6 ± 5.3 cm, VO2max 45.2 ± ml/min) underwent either a complex (the intermittent load type) or combined (the maximal strength and dynamic method) model of training. The training load was tailored to each athlete. Results showed that the complex model of training improved power (10 W/kg, p = 0.006) and height of vertical jump (5.3 cm, p = 0.001), weight of 1 Repeat Maximum (1RM) which was (5.8 kg, p = 0.015), power and speed in the acceleration phase of barbell half squats (BHS) at weights from 20 to 60 kg, and the number of repetitions in BHS (10.3%, p = 0.012). The combined model of training improved the time of shuttle run (0.44 s, p = 0.000), weight of 1RM in BHS (9.6kg, p = 0.000) and BP (4 kg, p = 0.000), power in the acceleration phase of BHS at weights from 50 to 60 kg, the number of repetitions in BP (14.3%, p = 0.000), BHS (9.4%, p = 0.002), barbell bench pulls (11.9%, p = 0.002) and sit-ups (7.7%, p = 0.001). These findings indicate that the complex model of training improves explosive abilities, whereas the combined model is effective for developing strength at weights close to players’ 1RM and for repeatedly overcoming resistance. Therefore, coaches should choose the training model based on the needs of individual players.

## 1. Introduction

Developing and loading strength abilities in football come from knowledge of game performance and its particular components and limitation factors [1]. Some of the most important factors in football are explosiveness activities such as changing direction, jumping, sprinting, and kicking [2]. Enhancements of these explosive performances have been found after strength training that improved the available force of muscular contraction in appropriate muscle groups. Vertical jump and sprint performances are connected with maximal muscular strength in football players [3,4,5,6]. Moreover, the heavy resistance training can also increase running economy and therefore aerobic performance [3,7,8].

Periodization of strength training mostly depends on the period of the season. Football players should achieve their peak as part of preparatory training and then maintain it for rest periods of up to 35 weeks [9]. Many authors [10,11,12] recommend using a traditional form with a progressive loading of training during the transition and preparatory period. A non-traditional form of periodization is used mostly during in-season and includes specific football exercises. 

Buzek [13] states that preparation games in football used to develop muscle strength are not sufficient for adequate adaptive changes. Universal strength training involves the development of maximum dynamic and explosive strength and can effectively take place in the gym. The intensity of this training should be very high, and it is not effective to carry out this exercise with football skills because it may restrict the exercise intensity and therefore the final changes will be smaller in the following developed indicators. Fitness trainers often use non-specific programs in football in order to achieve a higher level of adaptation, but, even outside of the field, an effort is still required to make exercise as specific and effective as possible for the demands of the sport. Authors [14,15,16] have revealed that the more similar a training exercise is to the actual physical performance, the greater the probability of transfer. Some controversy exists concerning the "transfer of training effect" from different methods of resistance-training programs to various athletic performance variables. A specific strength training method needs to be chosen based on the variables to be influenced. Knowledge of the effects of each method is crucial for the success of training with respect to physical and sports performance and in terms of preventing injuries [17]. Some studies [7,18], while monitoring relative muscle adaptation depending on the type of training, have shown that focusing on the ratio of the cross-section of muscle and neural adaptation provides the best impact velocity–force method, which focuses on explosiveness, maximum power and velocity strength. In contrast, for the hypertrophy of the muscle, the maximum power method is the best. Other authors [19,20,21] conducted research using various strength parameters and various types of training for strength-trained athletes. Their results confirmed that specific strength training stimuli evoke specific adaptation. Many studies have shown that the most effective method for explosive strength enhancement is the plyometric and weightlifting method consisting of maximal, submaximal and light weights as well as a combination of these methods [9,22,23,24,25,26,27]. However, the best gains in term of the physical markers were achieved by athletes who used a combined model approach. Specifically, Harris [19] exposed that the combination model of training was effective for improving most of the strength tests compared to models of training based on improving just high force or high power. The characteristics of the combination model are not the same in these studies, as they are a variable combination of different methods for developing strength abilities, using their advantages to achieve the best force gradient.

This study evaluates the effect of complex and combined models of traditional strength training on different strength abilities: maximal, endurance and explosive strength. The first model has been set up based on football actions in the game, basic principles of strength training and energy coverage of these actions. Players in the match need to use different kinds of strength abilities and muscle contractions and our aim was to train them based on the requirements of the game. Football actions during the game have a certain duration and intensity, and players overcome certain resistances. We wanted to incorporate these factors and imitate similar conditions in the strength training and determine if this kind of training with light weights and high volume would be effective. The second model was selected based on its effectiveness for developing several strength abilities. Comparison of the complex and combined models of the training could add valuable information about building and periodization of strength training in collective games. 

We hypothesized that the complex model of training improves explosiveness and endurance strength significantly more than the combined model, whereas maximal strength increases more after applying the combined model of training.

## 2. Materials and Methods

### 2.1. Participants

Thirteen professional soccer players (age 20.2 ± 3.3 years, body mass 57.2 ± 3.7 kg, height 163.6 ± 5.3 cm, VO2max 45.2 ± ml/min) volunteered to participate in the study. None of the participants had any musculoskeletal disorders, and all of them have played in the first national football league of Slovakia. The length of playing activity was 6.3 ± 1.9 years. Four players were in the junior national U19 team and six of them were in the senior national team. During the research period, it was recorded that all 13 players fully completed the study. At the beginning of the study 21 players were involved, but because of injury, sickness, and transfers to other clubs some of them did not finish both models of strength training. All 13 participants first completed the complex model and after one year in the same period completed the combined model of training. The procedures followed were in accordance with the ethical standards on human experimentation stated in compliance with the 1964 Helsinki Declaration and its later amendments. The project was approved by the ethics committee of the Faculty of Physical Education and Sports, Comenius University in Bratislava (1/0824/17 and 1/0089/20).

### 2.2. Training and Testing

Application of the complex and the combined model for the development of strength abilities took place during the transition period I (five weeks) and in the beginning of preparatory period II (four weeks). Both models were implemented for nine weeks, with 27 training units (TU) strength training, 13 TU football training and 2 TU aerobics. Duration of each TU was from 60 to 90 minutes (Table 1). Football training contained preparation games (7 vs. 7 up to 11 vs. 11).

Both strength models have the same principles: circuit organization form; three sets; 3 min rests between sets; and eight exercises which were focused on main muscle groups. Four exercises were always the same: sit-ups, barbell half squat (BHS), bench press (BP) and barbell bench pull (BBP). The next four exercises were performed with own weight and we used many of their variations based on the strength level of the players (plank, wall sit, push-ups, superman, etc.). These four exercises were performed on time (30s) or number of repetition (20×). All players had set up their individual weights on these exercises BHS, BP, BBP according to the input measurements of maximal strength 1RM at the beginning of the training program, and they increased their individual weight of resistance about 4%–6% every two weeks. The frequency of TU in both models was three times per week (Table 2).

The first experimental factor was the complex model of strength training, which is based on changing the contractions and varying the intensity during workout. Every football player started working out with a weight of 30% of their one repeat maximum (1RM). The weight was raised every two weeks by 4%–6% from 1RM until the end of the monitored period, so the players worked out the last week with 42%–45% of their 1RM. The speed of contractions was implemented in this order: the first 6 s were on velocity strength at maximum intensity; followed by isometric contractions in 4-second intervals in the middle position, endurance strength was applied for 40 s (without stopping movement and without emphasis on intensity of movements). The 4-second isometric contraction was then repeated and finally there were 16 s focusing on explosive strength (in their subjective maximum intensity during the concentric phase) in any number of repetitions (minimum one). Duration of one exercise was then 1 min 10 s. After the players finished the first exercise, they had 20 s to go to another station. In this circuit form they changed eight exercises in three sets with a rest interval of 3 min between the sets. This model simulated intermittent football actions in the football game (sprint–tackle–jogging–tackle–jump–rest). 

The second experimental factor was the combined model of strength training. This model was applied in the following season in the same period. In this stimulus was included two methods for developing strength abilities in one training unit (TU). The circuit form was used as in the previous year with the same eight exercises. In the first set, the maximal strength method was used with 3–6 repetitions, 80% from 1RM (concentric), with an interval of rest period between exercises which was under 20 s. In the second and third sets the dynamic method was used with 15–30 repetitions, with 30% of 1RM (concentric). The number of repetitions depended on the players’ subjective feelings. If they felt that the speed of the concentric phase was decreasing because of tiredness of muscles, then they were instructed to stop and move to the next exercise. In both these methods, players tried to hold maximal intensity in the concentric phase of movements. The load was raised every two weeks in the same way as in the complex model about 4%–6% from 1RM (concentric) from the input maximal strength test. Players worked out three times per week with a progression up to 92%–95% from 1RM in the first set and up to 42%–45% in the second and third set. This model was chosen based on its high effectiveness in a previous study [19].

### 2.3. Characteristics of Selected Tests and their Justification of Selection

Selected motor tests: Standing long jumpShuttle run (10 × 5 m)Vertical jumps (Jumper)

Players went through general warm up (10’) and then performed one standard standing long jump (SLJ) to determine the explosive power of their lower limbs. After 5 mins of break, this was followed by one set of 10 s jumping, continued on the FiTRO Jumper device (FITRONIC, Bratislava, Slovak Republic). The procedure was jumping as high as possible (without using arms and bending knees) and staying on the mat as short a time as possible. The three highest jumps were selected and their mean results calculated [28,29]. The level of jumping abilities was measured with the FiTRO Jumper device, consisting of a contact mat placed on the floor and connected with an interface to the computer. The criterion of level evaluation of jumping abilities was the height of the jump (h) in cm with accuracy 0.1 cm and power (P) in (W/kg). The FiTRO Jumper device (FITRONIC, Bratislava, Slovak Republic) uses the relationship h = (g x Tf2) / 8 for measurement of jump height. The producer of FITRONIC s.r.o. guarantees accuracy and reliability of the device with certified simultaneous measurements with a spring mat from the company KISTLER [30]. For assessment of speed and agility, the shuttle run test (10 × 5m) was applied and the time was measured with the FITRO Gates photocell device (FITRONIC, Bratislava, Slovak Republic). The methodology of standing long jump and shuttle run was taken with the Eurofit fitness testing battery [31].

Interval control exercise:Barbell bench pulls (2 × 20 s) 16 kgSit-ups (2 × 20 s)Barbell half squats (2 × 20 s) 20 kgBench presses (2 × 20 s) 16 kg

After the shuttle-run test, players executed interval control exercises where an effort was made to achieve as many repetitions (n) as possible during 2 × 20 s periods with a passive rest of 20 s. The players always started from the low position, except for in the barbell half squat, and also rested in the same position between two intervals of loads. It was necessary to execute these exercises with a full range of motion. The weight of the barbell on the pretest and post-test was the same. These interval control exercises measure the strength and endurance capacity of muscles.

Tests for determining maximal performance and maximal strength:

The players went through a general warm up (5’), after which followed a stepwise diagnostic series for detecting their one repeat maximum (1RM) with the following exercises: barbell half squats (BHS), bench press (BP) and barbell bench pull (BBP). The players did this series with an effort to achieve a maximal speed in the concentric phase with different weights from 20 kg up. After every successful attempt an increase in weight of about 5 kg followed. All strength ability parameters were registered until the 1RM was found [32].

For gaining the strength ability factors a monitoring machine, FITROdyne Premium (FITRONIC, Bratislava, Slovak Republic) was used. This device can record the vertical velocity and length of motion. This pc-based system, FiTROdyne, can capture biomechanical parameters during workout (FiTRONiC, SVK). The device contains a sensor unit with a precise encoder and a reel. While dragging the tether (which is attached by means of a hook to the bar axis) the reel rotates and computes speed. Information from a sensor is transferred to the PC by a USB cable. The process works on Newton’s second law of motion and Newton’s law of gravitation constant. The force is calculated for moving a bar in a vertical direction as the sum of gravitational force and acceleration force. Acceleration of vertical movements is gained by derivation of vertical velocity. Power is measured as a result of force and velocity and the actual position by integration of speed. Extensive computer software permitted the collection, calculation, and online display of the elementary biomechanical parameters involved in the workout. Previous studies have acknowledged that measurements of peak and mean power during resistance exercises using the FiTRO Dyne Premium system provide reliable data [33,34].

### 2.4. Statistical Analyses

Data is presented as means and standard deviations (SD). The Shapiro–Wilk Test for normality was performed on all variables. Data showed a normal distribution. A paired Student’s T Test was employed to find the significant differences from pre to post- tests in complex model 1 and combined model 2. For determination of changes between the two models of training, two-way repeated measures ANOVA and a Bonferroni post-hoc test were used. The level of significance was set at p *≤* 0.05 and statistically significant differences were marked with a symbol (*). In order to interpret the practical significance of the research results, the effect size (ES) was reported. Cohen [35] defined effect sizes as “small, d= 0.2,” “medium, d = 0.5,” and “large, d= 0.8”. Statistical and data analyses were performed using the statistical program IBM SPSS Statistics 21 (IBM Corporation, Armonk, NY, USA).

## 3. Results

### 3.1. Changes in Standing Long Jump and Shuttle Run (10 × 5m) after Complex and Combined Models of Training

The combined model players significantly decreased the time of the shuttle run (p = 0.000, d = 0.91) but did not significantly change the length of the standing long jump. The complex model of training failed to improve the standing long jump length and the time in the shuttle run test (Table 3).

### 3.2. Changes in Power and Height of Vertical Jump after Complex and Combined Models of Training

After using the complex model of training, the power (P) and height of jump (h) increased significantly. On the other hand, their values significantly decreased after the combined model of training.

The best values of power in the concentric phase of take-off and height of the jump were achieved during the pretest measurement of the second model, while the lowest values were observed during the pretest measurement of the first model. Power increased by 10 W/kg (p = 0.006, d = 1.16) and jump height about 5.3 cm (p = 0.001, d = 1.17) at the post-test of model 1. However, the power decreased about 7.8 W/kg (p = 0.038, d = 0.038) and jump height about 2.9 cm (p = 0.036, d = 0.036) in the post-test model 2 (Table 4).

### 3.3. Changes in Weight in the Diagnostic Series (1RM - one Repeat Maximum) after Complex and Combined Models of Training

The weight during the half squat with a barbell on the top of the back increased after both the combined models of training by 9.6 kg (p = 0.000, d = 0.85), and the complex model of training by 5.8 kg (p = 0.015, d = 0.63 ). The lifting weight also increased during the bench press after the second model of training by 4 kg (p = 0.000, d = 0.84), whereas no significant changes were found in the weight of 1RM BBP (Table 5).

### 3.4. Changes in Power and Speed Produced during Half Squat at Weights from 20 kg to 60 kg after Complex and Combined Models of Training

The mean speed and power in the concentric phase of BHS increased (10.8 cm/s and 26.5 W respectively, p = 0.001, d = 0.48) after the complex model of training (Figure 1). Their values increased (7 cm/s and 24.4 W respectively, p = 0.005, d = 0.36) also after the combined model of training (Figure 2). The mean power increased with increasing weights and began to decline at 40 kg during both pretest and post-test measurements in the complex model of training. The mean power and speed in the acceleration phase of BHS increased significantly at weights of 20kg (p = 0.03, d = 0.98), 25kg (p = 0.031, d = 0.67), 30kg (p = 0.003, d = 0.21), 35kg (p = 0.029, d = 0.63), 40kg (p = 0.003, d = 1.18), 45kg (p = 0.009, d = 1.05), and 60kg (p = 0.025, d = 1.45). 

Peak values of power were achieved at 45 kg prior to the training and increased to 55 kg after the combined model of training. The mean power and speed in the acceleration phase of BHS significantly increased at weights of 50 kg (p = 0.009, d = 0.86), 55 kg (p = 0.008, d = 1.29) and 60 kg (p = 0.033, d = 0.74).

### 3.5. Changes in Maximal Power in Diagnostic Series BHS after Complex and Combined Models of Training

Maximal values of power in a diagnostic series of BHS improved from 313W to 345W (p = 0.001, d = 0.79) after the complex model of training. Furthermore, the mean work increased from 229 J to 283J (p = 0.002, d = 1.26). Additionally, there was an increase in the weights, from 41 to 49 kg (p = 0.007, 0.82), and mean force, from 400 N to 483 N (p = 0.007, d = 0.87). Players reached their maximum values of power at 41 kg, which corresponded to 68.33% of their 1RM concentric contraction, prior to the training and at 49 kg, which corresponded to 74.24% their 1RM concentric contraction, after the complex model of training. The maximum power improved about 5.91% to higher values of 1RM. 

After the combined model of training, the maximum values of power increased from 322 to 348 W (p = 0.066, d = 0.53). This change, as well as the range and acceleration of movement, was not significant. However, force improved from 411 to 475N (p = 0.001, d = 0.78), weight from 42 to 48 kg (p = 0.001, d = 0.72) and work from 234 to 269 J (p = 0.011, d = 0.72). In contrast, the speed of movement decreased from 79 to 74 cm/s (p = 0.017, 0.59). The maximum power was achieved at a weight of 42kg which corresponded to 71.2% 1RM concentric contraction, prior to the training, and increased to 48kg, which corresponded to 69.6% of 1RM, after the second model of training. The maximum power decreased about 1.63% to lower values from 1RM (Table 6).

### 3.6. Changes in the Performance of Interval Control Exercises after Complex and Combined Models of Training

After applying the combined model, the players improved in barbell bench pulls about five repetitions (p = 0.002, d = 0.77), sit-ups about three repetitions (p = 0.001, d = 0.90), barbell half squats about three repetitions (p = 0.002, 0.72) and bench press about five repetitions (p = 0.000, d = 0.86). After using the complex model of training, players improved just in barbell half squats about three repetitions (p = 0.012, d = 0.68); no significant changes were observed in the amount of repetitions in bench press, barbell bench pull or sit-ups (Table 7).

## 4. Discussion

The results of this study supported the hypothesis that both models would significantly improve performance in selected motor tests. While the complex model of training increased vertical jump height, the combined model of training decreased the time in the shuttle run. In the maximum concentric contraction, significant improvements were noticed after both models of training in the BHS, and combined model also improved the BP exercise. Both models were effective in improving different factors in the maximal power in BHS. Players improved significantly in power, weight, force and work after the first model as well as in weight, force and work after the second model of training. Players also achieved significant changes in power and speed during diagnostic series (20–60kg) on lower weights after the complex model of training in contrast to the combined model when they achieved significant results on heavy weights. The results from the post-tests of interval control exercises do not fully support the hypothesis that the complex model of training significantly improves more strength endurance abilities. In the combined model training, the players significantly improved in all exercises. The complex model positively influenced only exercise BHS.

It should be noted that, in football, the gains in muscular strength should not compromise velocity of movement. The aim of strength training is to enhance muscle strength so that acceleration and speed in football—specific movements, such as sprinting, turning and changing direction—may be improved [36,37].

The results from 1RM of exercise BHS do not fully correspond with the results of other authors [5,6,32,38,39]. They reported high correlations between muscular strength and explosive performance in the vertical jump test, because in the combined model we noticed significant improvement of 1RM in BHS but also a significant decline of power and height in the vertical jump test (p *≤* 0.05). This could have happened because of a different methodology of evaluation during the vertical jump test or inefficient activation of the stretch shortening cycle during this period [40]. Results from the complex model of the training support previous studies. 

Harris [19] applied the combined model of the training with a comparable length and similar amount of training (9 weeks, 4× per week). Subjects who trained based on the combined model improved significantly in more variables than groups who used high force or high-power strength training. As in the current study, subjects improved in 1RM squat, shuttle run, maximal strength of back muscles but showed no improvement in SLJ. The muscle strength and power enhancement in the combined model also confirm the other studies [2,10,19,26,41,42] showing that strength is enhanced after a short heavy training program. On the other hand, the stimuli (light weight, change of intensity, long duration) in the complex model also led to significant increase in explosiveness. 

Better results in maximal strength are commonly linked with improvements in relative strength and in power-related abilities [42,43,44]. Traditional strength exercises increase the maximum strength levels (one-repetition maximum, 1RM) of the lower body in football players [37,42]. These claims were confirmed in this study after applying both models of training in BHS. Players’ values of maximum power output went from 68.3 to 74.2% of 1RM after applying the complex model of training, compared to in the combined model from 71.2 to 69.6%. This mean percentage of maximal power from 1RM is higher compared to other studies [45,46], which recorded values around 40%–60% of 1RM. This could have happened because of players’ insufficient experience working out with heavy weights and therefore achieving poorer results in the maximum strength test and producing maximum power close to their 1RM. The results also show that the complex model could better influence maximal power output in the BHS than the combined model. This could influence the duration of exercises with changing types of contraction, mostly affecting the timing of neural stimulus, neuromuscular fatigue and motor unit firing adaptation. More investigation is needed to confirm this claim.

The main limiting factors of this study are that both interventions were done on one small group of players (without selections and randomizations and a control group). The effectiveness of the second model could also have been influenced by experiences from the previous year, and because three players have not had any experience with working out in the gym before the first model was applied. For more precise diagnostic results, it is recommended doing the control exercise (2 × 20s) with weights derived from a certain percentage of the maximum possible overcoming resistance 1RM. Our results confirm positive effects of both models of training. However, further research may be focused on adjusting entry training loads about 10%–15% from 1RM in both models or change time intervals in the complex model of training.

## 5. Conclusions

These findings demonstrate the positive effect of both complex and combined models of training on the strength abilities of football players. However, each model of training has a different impact on the development of muscle power and strength. The new approach in the complex model of training was effective in improving explosiveness and producing power during exercises and could be used for improvement of specific strength abilities in football. The combined model was mostly effective for developing strength in exercises at resistance close to 1RM, which could be important for players as a precondition for future achievement of a high level of explosiveness. Each model of training should be chosen accordingly, based on the needs of a group or of individual athletes. 

### Practical Application

For players who started in the gym or in the first phase of preparation, it would be better starting with the complex method of training to maintain good habits during their workouts and also because this type of training, based on our results, was more specific to football players. Furthermore, it can prepare individuals for the second stage of booster blocks, and, subsequently, use (either in one preparation period or in the following year) of the combined model of training, which will ensure the necessary development of maximum power parameters and thus create the necessary basis to possibly transfer to specific activities.

For players who have a well-mastered technique of exercises but without the necessary strength base, it is recommended to start with the combination model of training. In this example, it would be used to develop a sufficiently fundamental strength level and then finish with the complex model of training. Based on the order of the preparatory period, it would move from general to specific exercises.

For the third group of players, who have a sufficiently developed strength base and correct technique for executing individual exercises, it is recommend to work mainly on developing limiting factors during a strength preparation period and, thus, use the complex model of training.

## Figures and Tables

**Figure 1 sports-08-00042-f001:**
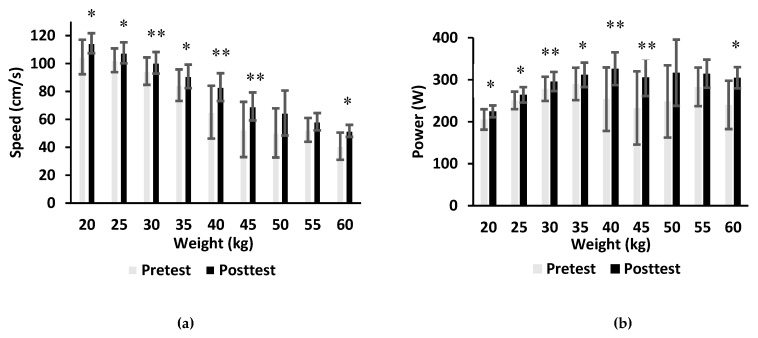
The mean velocity (**a**) and mean power (**b**) produced during barbell half squats at weights from 20 to 60 kg prior to and after 9 weeks of the complex model of training (*p ≤ 0.05, **p ≤ 0.01).

**Figure 2 sports-08-00042-f002:**
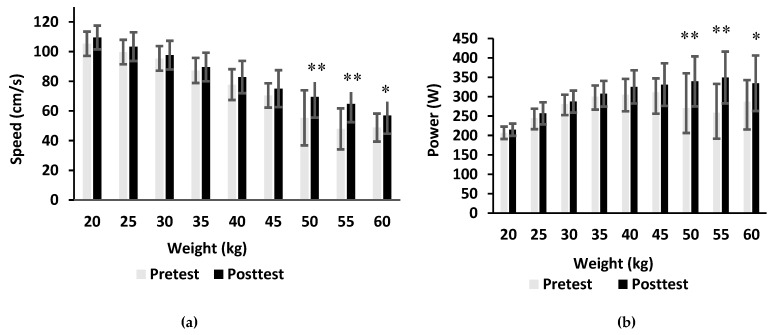
The mean velocity (**a**) and mean power (**b**) produced during barbell half squats at weights from 20 to 60 kg prior to and after 9 weeks of the combined model of training (*p ≤ 0.05, **p ≤ 0.01).

**Table 1 sports-08-00042-t001:** Training plans during the complex and combined models.

**Transition Period I—5 Weeks**						
**Monday**	**Tuesday**	**Wednesday**	**Thursday**	**Friday**	**Saturday**	**Sunday**
Strength Training	No Training	Strength Training	No Training	Strength Training	Football Training	No Training
75 min.	-	75 min.	-	75 min.	90 min.	-
**Preparatory period II—4 Weeks**						
**Monday**	**Tuesday**	**Wednesday**	**Thursday**	**Friday**	**Saturday**	**Sunday**
Strength Training	Football Training	Strength Training	Free / Aerobic	Strength Training	Football Training	No Training
75 min.	90 min.	75 min.	60 min.	75 min.	90 min.	-

**Table 2 sports-08-00042-t002:** Intensification and organization the during complex and combined models.

Models	Sets	Reps	Intensity	1–2 Week	3–4 Week	5–6 Week	7–9 Week	Rest
Model 1	1.–3.	Undefined	IM	30% 1RM	30% 1RM	35% 1RM	40% 1RM	3 min
Model 2	1.	3-6×	SM	80% 1RM	80% 1RM	85% 1RM	90% 1RM	3 min
	2.–3.	15-30×	SM	30% 1RM	30% 1RM	35% 1RM	40% 1RM	3 min

1RM = 1 Repeat Maximum, IM = Intermittent, SM = Subjective maximum.

**Table 3 sports-08-00042-t003:** Changes of selected motor tests after complex and combined models of training.

Exercise:	Pretest Model 1	Post-test Model 1	Pretest Model 2	Post-test Model 2
SLJ (cm)	195 ± 12	196 ± 9.4	198 ± 10.9	199 ± 9.1
10x5m (s)	18.22 ± 0.56	18.47 ± 0.43	18.49 ± 0.51	18.05 ± 0.46 **

10 × 5m = Shuttle run, SLJ = Standing long jump, ** p ≤ 0.01.

**Table 4 sports-08-00042-t004:** Changes in vertical jump after complex and combined models of training.

Models	Pretest P (W/kg)	Post-test P (W/kg)	Pretest h (cm)	Post-test h (cm)
**Model 1**	45.9 ± 8.7	55.9 ± 8.5 **	29.4 ± 3.9	34.7± 5.1 **
**Model 2**	59.7 ± 10.6	51.9 ± 8 *	36.7 ± 4.9	33.8 ± 4.6 *

h = Height, P = Power, * p ≤ 0.05, ** p ≤ 0.01.

**Table 5 sports-08-00042-t005:** Changes in mean weight in one repeat maximum.

Exercise	Pretest Model 1	Post-test Model 1	Pretest Model 2	Post-test Model 2
Barbell Half Squat	60 ± 8.55	65.8 ± 9.77 *	59.2 ± 10.35	68.8 ± 12.11 **
Bench Press	34 ± 5.83	34 ± 4.74	34 ± 4.87	38 ± 4.65 **
Barbell Bench Pull	35 ± 6.03	37 ± 5.41	37 ± 4.62	38 ± 4.65

* p ≤ 0.05, ** p ≤ 0.01.

**Table 6 sports-08-00042-t006:** Pre–post training changes in diagnostic series in the test barbell half squats.

	**Weight (kg)**		**Speed (cm/s)**	
	**Pretest**	**Post-test**	**Pretest**	**Post-test**
Model 1	41 ± 10	49 ± 9.6 **	81 ± 14.4	73 ± 13.4
Model 2	42 ± 6.7	48 ± 9.7 **	79 ± 8.9	74 ± 8 *
	**Power (W)**		**Range (cm)**	
	**Pretest**	**Post-test**	**Pretest**	**Post-test**
Model 1	313 ± 37.8	345 ± 42.8 **	59 ± 8.4	59 ± 6.7
Model 2	322 ± 41	348 ± 55.6	57 ± 4.4	57 ± 6.4
	**Force (N)**		**Work (J)**	
	**Pretest**	**Post-test**	**Pretest**	**Post-test**
Model 1	400 ± 97.8	483 ± 93.9 **	229 ± 41.7	283 ± 43.9 **
Model 2	411 ± 65.4	475 ± 95 **	234 ± 33.6	269 ± 59.9 **

* p ≤ 0.05, ** p ≤ 0.01.

**Table 7 sports-08-00042-t007:** Pre–post training changes in strength endurance in control exercises.

**Barbell Bench Pulls (n)**				**Sit-ups (n)**			
**Pretest Model 1**	**Post-test Model 1**	**Pretest Model 2**	**Post-test Model 2**	**Pretest Model 1**	**Post-test Model 1**	**Pretest Model 2**	**Post-test Model 2**
41 ± 4.1	44 ± 5.7	42 ± 7.5	47 ± 5.3 **	39 ± 4.6	41 ± 4.1	39 ± 3.3	42 ± 3.4 **
**Barbell half Squats (n)**				**Bench Press (n)**			
**Pretest Model 1**	**Post-test Model 1**	**Pretest Model 2**	**Post-test Model 2**	**Pretest Model 1**	**Post-test Model 1**	**Pretest Model 2**	**Post-test Model 2**
29 ± 5	32 ± 3.8 *	32 ± 4.4	35 ± 3.8 **	33 ± 7.1	36 ± 4.7	35 ± 5.8	40 ± 5.8 **

n = Number of repetitions, * p ≤ 0.05, ** p ≤ 0.01.

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
