# Peer review of "Effect of Two Strength Training Models on Muscle Power and Strength in Elite Women’s Football Players"

_sports, 2020, doi:10.3390/sports8040042_

Round 1
Reviewer 1 Report
I read with interest the manuscript by Pacholek & Zemkova titled, " Effect of two strength training models on muscle power and strength in elite women’s football players." The authors used 13 womens athletes to investigate the effect of 2 different models of training. The concept and research question is interesting, however, the methods and results do not support the conclusions, limitations are not addressed or acknowledged and the statistical approach may not be appropriate. The authors should be commended for their hard work and unique approach to this line of research. I am hopeful these comments are well-received and I look forward to seeing an improved version of this research in the literature.
Major comments:
Methods: The 2 models are not described well. Please describe all exercises, including total time of exercise per session and all movement performed for each. Specifically, highlight differences between the 2 training models.
Why was nonparametric testing used? Data should be first assessed for normality (consider shapiro-wilk test). Then appropriate statistical tests can be applied.
No subject characteristics. The participants are described as elite athletes, but there is no reported measure of aerobic fitness to verify this (e.g. VO2 max test). If vo2max test is not available, surrogate or predictive measure should be reported. Please refer to the exercise testing literature for viable options.
Please report basic subject characteristics as they changed over the study. These should be reported before and after each training period.
Please describe the athlete’s activity during their sports season and off season. i.e. what is happening with them outside of this training program. How much time was between the 2 models of this training program?
The training programs were not conducted in blind or randomized fashion. This needs to be presented as a limitation and discussed.
Training status of athletes was reported but not discussed.
Methods and results of outcomes measures were well described and presented with the exception that table titles should be on top and table legends (any abbreviations, exercise descriptions, statistics descriptions) should be on the bottom or at least not part of the title.
These results need to be placed in the context of the literature – have these models been compared before? What studies have reported them? How do your methods differ from theirs? Dif study population? Etc.
Minor comments:
Introduction is difficult to read and does not set up the research question well. Please rework to describe why these two models are selected and how they test the scientific question.
Reporting of methods and statistics is not standard. It is recommended the authors refer to similar literature in exercise testing for improved reporting. (e.g. it is not appropriate to describe p=.00; there should always be a leading zero before a decimal. If the value is less than p=0.01, it is reported as p<0.01, not p=.00)
Lines 352-380 are speculative and not supported by the data. Please keep discussion to topics related to measurable results.
Conclusion section is far too long and reads more like a results sections. Please rework. See other publications in sports performance as examples. Recommend work by John Hawley, Louise Burke, John Kirwan.
Reviewer 2 Report
Dear authors,
I believe that this study has got very interest in Sport Science, so it´s very limited the numbers of researches that have analysed the effect of a resistance training model of periodization in sport performance in elite women athlethes of this modality. So, I think that this study is very novelty. However, the statistical analysis is incorrect. It´s necessary to perform a new statistical analysis. Additionally, I want to indicate some considerations:
- I propose that keywords don´t be included in the title of the article for increasing the visibility of the paper in the databases.
- In the introduction is missing the possible effect of strength in economy and improvement of endurance performance in sport teams.
- It´s necessary to specify more about the model of strength periodization in the introduction.
- It´s necessary to include a table with the evolution of the loading of training during the intervention. Actually it´s very confused this section.
- Related to the jump test: was selected CMJ test? I´m not sure about the correct execution of the test.
- Related to the shuttle run, what was the system used for measure the time for covering the test?
- Related to the interval control exercise, have been employed this test previously in other studies?
- The statistical analysis is wrong. It´s necessary that author perform a two way of repeated measures ANOVA.
- The discussion must be revised attending to the results of the new analysis.
Reviewer 3 Report
Congratulations to the authors. This is an interesting study and the results are interesting. The objective of the article to evaluates changes in power and strength after two different models of 9-week strength training in elite women’s football players.
You have an interesting study, but your manuscript is difficult to understand and has errors that need clarification or revision. I offer the following suggestions for improvement.
I present the following in the order which the paper was written
Abstract
- The abstract should show the characteristics of the subjects. The method section should be more developed, indicating the main measurements that have been carried out. In addition, you have excessive description of your methods and You need to report your principal data here so readers can evaluate your results.
Introduction
Line 74-77: This should be better explained. Please amend it.
Line 90-93: Here the research hypothesis should be established. In this case, it seems that this is a conclusion. Please check it.
Participants
- It's probably best to say: 13 professional soccer players (age 20.2 ± 3.3 years, body mass 57.7 ± 4.7 kg, 96 height 164 ± 5.3 cm) volunteered to participate in the study. None of the participants had any musculoskeletal disorder and all of them have played in the first national football league of (indicate country).
- It should be indicated if the 13 participants were randomly assigned to different training groups or, conversely, if they first performed all the complex model and then the combination model (randomized and counterbalanced cross design).
Training and Testing
- When the authors refer to the TU, are they referring to a training session?.
- The authors establish "Every football player worked out with their 118 weight of 30% repeat maximum (1RM) according to the input measurements". However, there is no information on how the 1RM of the basic exercises was obtained. This information should be included.
- Line 120-122: “The football players 120 worked out with 42% from 1RM from the input measurements until the end of the 121 monitored period”. The 42% refers to the average training load of 9 weeks? or that they finished in week number 9 with an intensity of 42% of 1RM? I think I remember that they started with 30% ... Please clarify this.
- Line 125-127: “Then endurance strength 125 was applied for 40 seconds (the load without affecting the movement but without the 126 emphasis on intensity of movements)”. Were there any number of repetitions established? Or was it free for each participant?
- It could be interesting to create a figure with the training circuit, in which you can visualize the exercises and the main rest time between sets.
- Line 133-134: How much rest time was there between training sessions?
Line 141-143: “until 142 the players did not have any subjective feelings that their velocity was decreasing”. What do you mean? Muscle failure? or Until the person who controls the training decides that the exercise is not executing correctly?
Line 147-148: Again, do these percentages refer to the average training load?
Tests for determining maximal performance and maximal strength
- The percentages of 1RM used for training were used in this test? Please clarify it.
Statistical Analyses
- Line 216: In the methods section it should be clear which are the Model 1 and Model 2 for better identification in the results
Results
- All abbreviations that appear after the method section must be described in the method section or the first time they appear.
- Table 3: If the authors tried to put all the rows in order it would be easier to read.
- Line 281: “Changes in maximal power in diagnostic series after complex and combined models of training”. I don't know what the authors refer to with "diagnostic series". I don't know where this data comes from since there is no section with that name in the method. I think in the manuscript there is a problem with terminology.
- Line 312-313: This would be better placed in the conclusion.
Discussion
- The discussion should be reviewed.
- Line 346-352: Here the physiological or neuromuscular mechanisms that cause the observed changes should be exposed. It is not enough to indicate the changes observed and which studies agree with them.
- Line 359-375: This seems to be practical applications that should be placed in another section.
Conclusion
- Line 385-400: Honestly, this should be eliminated. This only the data that has been presented in the result section and in a part of the discussion. The conclusion serves to summarize the main idea that the authors want to convey and propose practical applications of their research.
- Line 401-410: This is a good conclusion. It would be interesting to add several of the previous practical applications.
Round 2
Reviewer 1 Report
The authors have appropriately addressed my concerns.
The statistics section reads a little odd. Please double check for english/typical reporting style. e.g. "2-way repeated measures ANOVA" or "paired Student's t-test."
No additional revisions/comments.
Author Response
Dear Reviewer,
- The statistics section reads a little odd. Please double check for english/typical reporting style. e.g. "2-way repeated measures ANOVA" or "paired Student's t-test."
We have changed the statistics section. Hopefully, it is more appropriate now.
Lines 255-265
Reviewer 2 Report
Dear authors,
I want to thank you because you have attending to all my comments. Sincerely, I believe that the quality of the paper have improved significantly, specially relative to the methods (actually, it´s possible to reproduce the research). I consider that the novelty and interest of the paper is high. Nevertheless, I think that the two-way ANOVA need to be improved. So, I want to share with you a table that I have created for you. I propose to adjust the results to this format. Also, authors need to remember perform Bonferroni like post-hoc. Additionally, in the text when authors write the statistical significant differences in the text, it could be suitable to perform the ES.
I think that with this new analysis could be improve significantly the quality of the article and it will be published.
|
Exercise |
Model |
Time |
p-value model |
p-value Time |
p-value model·time |
|
|
Pre |
Post |
|||||
|
SLJ (cm) |
Model 1 |
± |
± |
|
|
|
|
Model 2 |
± |
± |
||||
|
10 x 5 m (s) |
Model 1 |
± |
± |
|
|
|
|
Model 2 |
± |
± |
||||
|
CMJ (cm) |
Model 1 |
± |
± |
|
|
|
|
Model 2 |
± |
± |
||||
|
CMJ (W/kg) |
Model 1 |
± |
± |
|
|
|
|
Model 2 |
± |
± |
||||
|
Barbell Half Squat |
Model 1 |
± |
± |
|
|
|
|
Model 2 |
± |
± |
||||
|
Bench Press |
Model 1 |
± |
± |
|
|
|
|
Model 2 |
± |
± |
||||
|
Barbell Bench Pull |
Model 1 |
± |
± |
|
|
|
|
Model 2 |
± |
± |
||||
Author Response
Dear Reviewer,
- Nevertheless, I think that the two-way ANOVA need to be improved. So, I want to share with you a table that I have created for you. I propose to adjust the results to this format. Also, authors need to remember perform Bonferroni like post-hoc. Additionally, in the text when authors write the statistical significant differences in the text, it could be suitable to perform the ES.
Thank you very much for your points. We have add to our results: effect size values for all significant changes and also a Bonferroni like post-hoc test was performed. The t-values are already in the text and for this reason we haven’t put them to the table even though your table is much more organized than ours.
We really appreciate your valuable advice, effort and time, which you spent with this manuscript…. Thank you very much again.
Reviewer 3 Report
I appreciate the efforts made by the authors in order to improve their paper. Most of my concerns/comments have been considered adequately. In addition, I believe that the new conclusions and practical applications are a very strong point of this article so that the scientists / trainers know which methodology to use according to their needs. Only there are still minor comments remaining:
- On page number 4, the new tables have the same numbering (Table 3), but I think they are Table 1 and Table 2.
- I don't know if the limitations of the study should go at the end of the discussion or in a separate section. It could be more interesting not to remove potential from the discussion. But it is just an idea.
Author Response
Dear Reviewer,
- On page number 4, the new tables have the same numbering (Table 3), but I think they are Table 1 and Table 2.
Thank you that you pointed the numbering of tables. We have repaired it.
- - I don't know if the limitations of the study should go at the end of the discussion or in a separate section. It could be more interesting not to remove potential from the discussion. But it is just an idea.
Limitation of a study are "generally" placed either at the begging or end of the discussion but we don’t mind to put it in a separate section.
Thank you very much again for all your points and time.